# Reporter Flaviviruses as Tools to Demonstrate Homologous and Heterologous Superinfection Exclusion

**DOI:** 10.3390/v14071501

**Published:** 2022-07-08

**Authors:** Francisco J. Torres, Rhys Parry, Leon E. Hugo, Andrii Slonchak, Natalee D. Newton, Laura J. Vet, Naphak Modhiran, Brody Pullinger, Xiaohui Wang, James Potter, Clay Winterford, Jody Hobson-Peters, Roy A. Hall, Alexander A. Khromykh

**Affiliations:** 1School of Chemistry and Molecular Biosciences, The University of Queensland, Brisbane, QLD 4072, Australia; f.javiertorrestobar@uq.edu.au (F.J.T.); r.parry@uq.edu.au (R.P.); a.slonchak@uq.edu.au (A.S.); natalee.newton@uq.edu.au (N.D.N.); l.vet@uq.edu.au (L.J.V.); n.modhiran@uq.edu.au (N.M.); b.pullinger@uq.net.au (B.P.); xiaohui.wang@uq.edu.au (X.W.); j.potter@uq.edu.au (J.P.); j.peters2@uq.edu.au (J.H.-P.); 2Mosquito Control Laboratory, QIMR Berghofer Medical Research Institute, Brisbane, QLD 4006, Australia; leon.hugo@qimrberghofer.edu.au (L.E.H.); clayw@qimr.edu.au (C.W.); 3Australian Infectious Diseases Research Centre, Global Virus Network Centre of Excellence, Brisbane, QLD 4029, Australia

**Keywords:** Binjari virus, flavivirus, superinfection exclusion, insect-specific viruses, reporter viruses, CPER

## Abstract

Binjari virus (BinJV) is a lineage II or dual-host affiliated insect-specific flavivirus previously demonstrated as replication-deficient in vertebrate cells. Previous studies have shown that BinJV is tolerant to exchanging its structural proteins (prM-E) with pathogenic flaviviruses, making it a safe backbone for flavivirus vaccines. Here, we report generation by circular polymerase extension reaction of BinJV expressing zsGreen or mCherry fluorescent protein. Recovered BinJV reporter viruses grew to high titres (10^7−8^ FFU/mL) in *Aedes albopictus* C6/36 cells assayed using immunoplaque assays (iPA). We also demonstrate that BinJV reporters could be semi-quantified live in vitro using a fluorescence microplate reader with an observed linear correlation between quantified fluorescence of BinJV reporter virus-infected C6/36 cells and iPA-quantitated virus titres. The utility of the BinJV reporter viruses was then examined in homologous and heterologous superinfection exclusion assays. We demonstrate that primary infection of C6/36 cells with BinJV_zsGreen_ completely inhibits a secondary infection with homologous BinJV_mCherry_ or heterologous ZIKV_mCherry_ using fluorescence microscopy and virus quantitation by iPA. Finally, BinJV_zsGreen_ infections were examined in vivo by microinjection of *Aedes aegypti* with BinJV_zsGreen_. At seven days post-infection, a strong fluorescence in the vicinity of salivary glands was detected in frozen sections. This is the first report on the construction of reporter viruses for lineage II insect-specific flaviviruses and establishes a tractable system for exploring flavivirus superinfection exclusion in vitro and in vivo.

## 1. Introduction

The *Flavivirus* genus (family *Flaviviridae*) contains several medically important viruses, including yellow fever (YFV), dengue (DENV), and Zika (ZIKV), which are responsible for millions of global infections annually, resulting in a range of disease outcomes in humans [1,2]. These pathogenic flaviviruses exist in transmission cycles that primarily involve horizontal transmission between mosquitoes and amplifying vertebrate hosts.

Flavivirus genomes contain a positive-sense (+), single-stranded RNA molecule with a single open reading frame, encoding a polyprotein post-translationally cleaved into three structural (capsid (C), premembrane (prM), and envelope (E)) and seven non-structural (NS) proteins (NS1, NS2A, NS2B, NS3, NS4A, 2K, NS4B, and NS5) [3,4,5]. Additionally, the genome contains ~100 nt 5′ untranslated region UTR and ~400–700 nts 3′ untranslated regions facilitating viral RNA replication and translation [6].

In addition to vector-borne flaviviruses that can infect vertebrates (vertebrate-infecting flaviviruses, VIFs) and cause disease, several insect-specific flaviviruses (ISFs), named for their insect-restricted host range, have been isolated and characterised (reviewed by [7]). They belong to two phylogenetically distinct groupings: lineage I or classical ISFs (cISF), in which all members form a monophyletic clade, and a lineage II or dual-host-affiliated ISF (dISF) clade of flaviviruses that also cannot replicate in mammalian cells but are closely related phylogenetically to VIFs [7].

Given the close genetic relationship of VIFs to lineage II ISFs, reverse-genetics studies have demonstrated that some dISFs are tolerant of the exchange of structural protein genes (prM-E) with those of pathogenic vertebrate-infecting flaviviruses [8,9,10,11]. One lineage II ISF, Binjari virus (BinJV), first isolated from *Aedes (Ochlerotatus) normanensis* mosquitoes in Australia [8], is the backbone for numerous chimeric VIF BinJV vaccine candidates [12,13,14,15]. BinJV replication in vertebrate cells is restricted at multiple stages of cellular infection, including inefficient cell entry and susceptibility to antiviral responses triggered by the BinJV CpG genome composition [16,17].

In addition to using ISFs for safe flavivirus vaccine backbones, there has been a considerable examination of exploiting ISFs and other insect-specific viruses (ISVs) in biocontrol strategies to control vector-borne disease [18,19]. This is due to the observation that primary or co-infections of ISVs have been shown to restrict or modulate secondary flavivirus infections in vitro [20,21,22,23,24] and in vivo in mosquitoes [23,25].

Fluorescent imaging of viruses encoding fluorescent reporter groups (i.e., zsGreen, mCherry) enables examination of virus replication or virus protein localisation in living organisms. Furthermore, it provides a chance to investigate their real-time dynamics without needing expensive imaging reagents [26].

The first infectious flavivirus reporter virus was engineered for Japanese encephalitis virus (JEV) in 2003 and contained GFP and luciferase in the 3′ UTR [27]. Unfortunately, early reporter viruses were plagued with genetic instability and spontaneous loss of the insert (Reviewed by [28]). A YFV reporter virus described in 2007 [29] improved stability by fusing the first 101-aa capsid sequence with GFP, followed by a foot-and-mouth disease virus 2A ribosome stuttering peptide (FMDV 2A), codon-optimised capsid gene, and the rest of YFV polyprotein [29]. Since these early studies, many reporter viruses for the VIF flaviviruses have been constructed [28]. In contrast, only one report has demonstrated reporter viruses’ construction for the Lineage I ISF Niénokoué virus [30].

Here, we generated BinJV reporter viruses expressing the fluorescent proteins zsGreen or mCherry inserted between partial and complete capsid gene sequences using our established Circular Polymerase Extension Reaction (CPER) method [31,32]. The CPER method allows de novo generation of infectious DNA from viral cDNA fragments with 20 to 40-nt complementary overlapping ends and a linker fragment containing insect expression promoter.

We examine the replication kinetics of recovered BinJV reporters compared to WT BinJV in vitro in *Aedes albopictus* cells and in vivo via intrathoracic injection in *Aedes aegypti*. The BinJV reporter viruses were then utilised to explore the homologous exclusion phenomenon or heterologous exclusion with the ZIKV reporter virus [33].

## 2. Materials and Methods

### 2.1. Cell Culture

*Aedes albopictus* larvae cells (C6/36, ATCC–CRL-1660) were grown at 28 °C in Royal Park Memorial Institute (RPMI) medium (Gibco, Waltham, MA, USA), 10% fetal calf serum (FCS; Bovogen, Keilor East, Australia), and 1% GlutaMAX (200 mM; Gibco). African green monkey (*Chlorocebus* sp.) kidney cells (Vero76, ATCC–CRL-1587) were maintained at 37 °C in 5% CO_2_ in Dulbecco modified Eagle medium (DMEM; Gibco) supplemented with 10% FCS and 1% GlutaMAX (Gibco). For viral infections, cells were maintained in an FCS reduced media (2% FCS) with 10,000 U/mL of penicillin and streptomycin (Gibco).

### 2.2. Viral RNA Extraction and RT-PCR Amplification of Viral Genomic Fragments

Viral RNA from an early passage of the BinJV isolate BFTA20 (GenBank: MG587038) was isolated using TRIreagent (Sigma, Ronkonkoma, NY, USA) and used as a template for first-strand cDNA synthesis with SuperScript IV reverse transcriptase (Invitrogen, Waltham, MA, USA) as previously described [34]. After first-strand cDNA synthesis, RNA was removed from RNA–DNA duplexes by incubation of RT mixture with 1 μL of RNase H (NEB, Ipswich, MA, USA) and RNase A for 20 min at 37 °C and 20 min at 60 °C. Then, RNase-treated cDNA was used as a template for PCR to amplify two viral fragments encompassing the BinJV genome (F2 and F3) using PrimeStar GXL Polymerase (Takara, Shiga, Japan) and with the following primer pairs BinJV-2F: 5′-AGGAGCGATGGCTGCCACTTTACGCA-3′, BinJV-2R: 5′-CCACAACACAGTCCCCCGCTTGTTTGATTT-3′, BinJV-3F: 5′-AAATCAAACAAGCGGGGGACTGTGTTGTGG-3′, and BinJV-3R: 5′-GCGAGGAGGTGGAGATGCCATGCCGACCCAGATACTTGATGTTTCTCAATCCCCAATCCA-3′. The cycling conditions were Stage 1: 3 min (98 °C); Stage 2: 30 cycles of 10 s (98 °C), 15 s (58.3 °C), 6 min 30 s (68 °C); Stage 3: 5 min (68 °C). PCR products were separated in 1% agarose gel, and DNA was extracted from the gel using Monarch DNA Gel Extraction Kit (NEB).

### 2.3. Construction and Amplification of Reporter and Linker Fragments

The complete 96 nt BinJV 5′ UTR previously characterised [35], the first 153 bp of the BinJV capsid gene, the zsGreen or mCherry fluorescent protein gene, a foot-and-mouth disease virus (FMDV) 2A ribosome stuttering peptide sequence (LLNFDLLKLAGDVESNPG↓P), the full-length codon optimised BinJV capsid gene, and the 5′ end of BinJV prM gene were ordered as a gBlocks Gene Fragments (IDT, Singapore), blunt-end cloned into the pUC19 vector, and confirmed by Sanger sequencing. These plasmids were then used in PCR as templates with primers Reporter_F 5′-GCCTATAAATACAGCCCGCAACGATCTGGTAAACAGTATATTTTGCGTGTGCGTTTCAAA-3′ and Reporter_R 5′-TGCGTAAAGTGGCAGCCATCGCTCCT-3′ to yield 1412 and 1427bp fragments (F1) for zsGreen and mCherry reporter genes, respectively. An insect-optimised OpIE2-CA promoter-UTR linker fragment was amplified from the OpIE2-CA-UTR linker plasmid [11] with the following primer pair OpIE2_Link_F 5′-TGGATTGGGGATTGAGAAACATCAAGTATCTGGGTCGGCATGGCATCTCCACCTCCTCGC-3′ and OpIE2_Link_R 5′-TTTGAAACGCACACGCAAAATATACTGTTTACCAGATCGTTGCGGGCTGTATTTATAGGC-3′. The OpIE2-CA promoter-UTR linker fragment contains the last 20 nucleotides of BinJV 3′UT and the hepatitis delta virus ribozyme (HDVr) to generate authentic 3′-OH RNA. The linker also contains an SV40 polyA signal for transcription termination, a modified baculovirus promoter (OpIE2-CA) [11] for in vivo transcription of viral RNA by cellular RNA polymerase II, and the first 20 nucleotides of BinJV 5′ UTR.

### 2.4. CPER Assembly, Recovery of Reporter Viruses and RT-PCR of Viral RNA

The purified reporter fragment, two BinJV viral dsDNA fragments, and linker fragment containing overlapping sequences of 26–60 nucleotides were pooled in equimolar amounts (0.1 pM each) in a 50 µL reaction containing dNTPs (200 µM each) and 2μL of Prime Star GXL DNA polymerase in a GXL reaction buffer. The following cycling conditions were used: Stage 1: 98 °C (30 s), Stage 2: 12 cycles of 98 °C (10 s), 55 °C (20 s), and 68 °C (10 min). Infectious cDNA from CPER assembly was transfected into C6/36 cells using Effectene transfection reagent (QIAGEN, Hilden, Germany) according to the manufacturer’s instructions. To generate P_1_ stocks and confirm the reporter genes’ retention in the BinJV recombinants, cell culture fluid (P_0_) was harvested on day ten post-transfection and blind-passaged once onto C6/36 cells. At seven days post-infection, viral RNA was isolated, and cDNA was prepared as above. PCR was performed with PrimeStar GXL Polymerase (Takara, Japan) and with the following primer pairs: RGene_F: 5′-AAACTCAGGAGGCCCGTTAAACGGGCCGTC-3′ and RGene_R: 5′-GAGAACCCTCCGGGGTCCAGCGGC TCGAGGAACTG-3′. The cycling conditions were Stage 1: 3 min at 98 °C; Stage 2: 30 cycles 98 °C (10 s), 66.2 °C (15 s), 68 °C (1 min 30 s); Stage 3: 5 min at 68 °C. ZIKV_mCherry_ virus was also generated by CPER based on a published design [33].

### 2.5. Growth Kinetics and Immuno-Plaque Assay (iPA)

Growth kinetics of wild-type and reporter BinJV were assessed in C6/36 cells. Cells were seeded in six-well plates at 10^6^ cells per well and inoculated with BinJV_WT_, BinJV_mCherry_ or BinJV_zsGreen_ at a multiplicity of infection (MOI) of 0.1 for one hour. After removal of the inoculum, cells were washed three times and replaced with 2 mL of RPMI (2% FCS). Infected cells were incubated for seven days at 28 °C with culture fluid harvested at zero, three, five, and seven days post-infection. Viral titres of the harvested supernatant were quantitated with a foci-forming immunoplaque assay [34,36].

For virus immunoplaque assay (iPA), C6/36 cells (10^5^ cells per well) or Vero76 cells (2 × 10^4^ cells per well) were seeded and infected with serially diluted inoculum media. For BinJV and ZIKV titration in C6/36 cells, cells were fixed at 72 h post-infection. For ZIKV titration in Vero76, cells were fixed at 48 h. All cells were fixed with 100 μL/well of 80% acetone in PBS solution for 1 h at −20 °C and washed with PBS. Plates were dried and blocked for 30 min with 150 μL/well of Pierce™ Clear Milk blocking solution (ThermoFisher Scientific, Waltham, MA, USA). After blocking, plates were incubated with 50 μL/well of primary antibody for one hour. BinJV WT and reporter viruses were titred using a primary mouse anti-NS1 flavivirus antibody 4G4 [37] (1:100). Zika was titred using a primary human anti-ZIKV E protein antibody Z67 (1:5000) [38,39]. Recovered P_1_ ZIKV_mCherry_ used for super-exclusion experiments were titred with mouse 4G4 (1:100). Cells were subsequently incubated with 50 μL/well of goat anti-human IRDye 800CW secondary antibody (1:2000) or goat anti-mouse IRDye 800CW secondary antibody (1:2000) (LI-COR). All antibodies were diluted with Clear Milk blocking buffer (Pierce) in phosphate-buffered saline containing 0.05% Tween 20 (PBS-T). Primary and secondary incubations were performed at room temperature for one hour. After each incubation, plates were washed 5 × 5 min with PBS-T. Plates were then scanned using an Odyssey CLx Imaging System (42 μm; medium; 3.0 mm), foci-forming units (FFU) were counted using the Image Studio Lite software (v 5.2.5) (LI-COR, Lincoln, NE, USA), and titres were calculated as FFU per mL (FFU/mL).

### 2.6. Superinfection Exclusion Assays

C6/36 cells were seeded in triplicates in 24-well black, glass-bottom plates (Eppendorf) and infected at ~90% confluence with BinJV_zsGreen_ at a MOI = 1 or “mock-infected” with media only and incubated at 28 °C. Cells were observed daily for five days under green channel of an inverted epifluorescence microscope. Five days post-infection demonstrated the time when green fluorescence was predominant in cell monolayers. After five days, the growth medium was removed, and cells were washed three times with RPMI and infected with either BinJV_mCherry_ at an MOI of 1, or ZIKV_mCherry_ at an MOI of 1, or mock-infected. Finally, plates were incubated for another five days, with images of cells taken under green and red fluorescent light with the camera of the inverted epifluorescence microscope. In parallel, samples were taken immediately after infection with either homologous (BinJV_mCherry_) or heterologous (ZIKV_mCherry_) viruses for 0-time points and every day for five days. Culture supernatant from each experiment was taken daily.

### 2.7. zsGreen and mCherry Fluorescence Quantification by Microplate Reader

For the live quantification of BinJV virus reporters, 25 µL of serially diluted (10^−1−5^) culture fluid from infection trials and growth kinetics were infected into C6/36 cells seeded at 10^6^ cells per well in Black Nunc MicroWell 96-Well Optical-Bottom Plates (Thermo Scientific™ Catalogue #165305). Cells were inoculated for an hour and supplemented with 100 µL of RPMI culture medium with 2% FCS. Plates were scanned daily using the Varioskan™ LUX (Thermo Scientific) microplate reader. The bottom of the wells were analysed with the SkanIt™ Software v6.0.1 (ThermoFisher Scientific, Waltham, MA, USA) under the following conditions: 29 points per well, measurement time: 100 ms, excitation bandwidth: 5 nm and automatic dynamic range. For measurement of zsGreen fluorescence in cells, the excitation wavelength (λ_ex_) was 488 nm, and the emission wavelength (λ_em_) was 506 nm. For quantification of mCherry fluorescence in cells, the λ_ex_ was 585 nm and the λ_em_ 610 nm. For linear regression analysis, plates were read at five days post-infection.

### 2.8. Microinjection of Aedes aegypti

Wildtype *Aedes aegypti* (Cairns strain) used for infection have been maintained since 2015 in the QIMR Berghofer insectary at 28 °C with 70% relative humidity and 12:12 hr light cycling with dawn and dusk fading. Adults were maintained in 30 cm cages (BugDorm, MegaView Science Education Services Co., Ltd., Taichung, Taiwan). For colony maintenance, adults were fed defibrinated sheep blood (Serum Australis, Manila, NSW, Australia) weekly and continuously provided with 10% sucrose solution.

Adult mosquitoes (7–10 days old) were anaesthetised with CO_2_, placed on a petri dish on ice and microinjected with 200 nL of BinJV_zsGreen_ at 1.86 × 10^6^ FFU/mL (delivering ≈ 372 FFU/mosquito) using a Nanoject III programmable nanolitre injector (Drummond Scientific, Broomall, PA, USA) with pulled glass capillary tube needles. Injected mosquitoes were transferred to 750 mL cups with gauze lids and maintained at 28 °C, 75% relative humidity, 12:12 hr day: night light cycle inside a Fitoclima 1200 environmental chamber (Aralab, Rio de Mouro, Portugal) for 7 days post-injection.

### 2.9. Histological Analysis and Fluorescence Microscopy

For frozen sectioning, mosquitoes were embedded in Optimal Cutting Temperature Compound (OCT) and frozen in an ethanol and dry ice slurry. Mid-sagittal sections (12 µM) were cut using a cryostat microtome. Sections were stained in 0.14 µg/mL 4′,6-diamidino-2-phenylindole (DAPI) for 5 min and washed in tris-buffered saline-0.025% Tween 20. Microscopy was performed using an Aperio ScanScope fluorescent microscope using filters for DAPI and FITC with exposure times of 0.2 s and 0.125 s, respectively. In preliminary experiments, mosquito specimens were also fixed in 4% paraformaldehyde, 0.5% Triton X overnight, transferred to 70% ethanol, embedded in paraffin, and sectioned using standard procedures [40]. CPER-transfected and BinJV or ZIKV reporter virus-infected C6/36 cells were visualised and imaged using an inverted Nikon ECLIPSE Ts2 microscope.

### 2.10. Statistical Analysis

A two-way ANOVA test with Dunnett’s correction was used for growth kinetics to compare viruses at each time point to WT BinJV. A general linear analysis was used for Figure 2D,E. All data were analysed and visualised using GraphPad Prism software v9.0.0 (San Diego, CA, USA). The level of statistical significance was set at 95% (*p*  ≤  0.05).

## 3. Results

### 3.1. Construction and Generation of BinJV Expressing zsGreen and mCherry Viruses

The reporter gene insertion strategy, which permits the stable expression of fluorescent proteins in insect cells infected with flaviviruses [33,41,42], was employed to characterise BinJV properties. To this end, the CPER platform [8,16,31] was utilised to insert sequences of zsGreen or mCherry genes between the first 153 nucleotides of the BinJV capsid gene, containing cis-acting RNA elements essential for virus RNA replication [43,44,45] and the full-length BinJV capsid gene, with codon optimisation to prevent recombination and potential loss of reporter genes during virus replication (Figure 1A) [33]. Both reporters also contained an FMDV 2A ribosomal skipping sequence at the C-terminus to ensure the release of reporter proteins from the downstream viral polyprotein (Figure 1A). PCR amplification and electrophoresis of the two reporter gene fragments (F1) along with two additional fragments encompassing the BinJV genome (F2 and F3) and the insect optimised OpIE2-CA promoter-UTR linker fragment showed correct sizes of the five CPER fragments (Figure 1B).

Amplicons were pooled in equimolar amounts and circularised using CPER assembly using GXL DNA polymerase. The CPER was then transfected into C6/36 cells and visualised daily from two days post-transfection over nine days with an inverted epifluorescence microscope (Figure 1C). We observed the signal of zsGreen positive cells in the green channel at four days post-transfection and the mCherry signal in the red channel at five days post-transfection. Both transfected cell lines showed increased signal from day six to day ten when the supernatant containing the infectious virus was harvested. No autofluorescence of mock-transfected (media only) cells was observed.

The harvested culture supernatant containing passage 0 (P_0_) BinJV reporter viruses was then passaged in C6/36 cells for seven days. The resultant passage 1 (P_1_) stock was quantitated using a pan flavivirus antibody 4G4 by iPA. The titres were 1.86 × 10^6^ FFU/mL for BinJV_zsGreen_ virus and 1.14 × 10^6^ FFU/mL for the BinJV_mCherry_ virus. The P_1_ BinJV reporter viruses were also examined for the retention of the reporter gene insert by RT-PCR, with the reporter genes shown to be retained in the P_1_ viruses (Figure 1D).

### 3.2. Growth Kinetics and In-Cell Fluorescence Quantification of BinJV Reporter Viruses

To examine the kinetics of the recovered BinJV reporters compared to CPER generated WT BinJV [35], *Aedes albopictus* C6/36 cells were infected at an MOI of 0.1 with P_1_ virus stocks. Culture supernatants were harvested at days zero, three, five, and seven after infection and titred by iPA using the 4G4 antibody (Figure 2A).

For BinJV_zsGreen,_ the virus steadily increased to mean final titres of 4.27 × 10^8^ FFU/mL at seven days from 2.48 × 10^8^ FFU/mL and 2.81 × 10^8^ FFU/mL at three and five days, respectively (Figure 2A). For BinJV_mCherry_, replication peaked at three days post-infection for a mean titre of 5.57 × 10^7^ FFU/mL and declined to 4.37 × 10^7^ FFU/mL on day five with final titres of 6.17 × 10^7^ FFU/mL at seven days. Conversely, the WT BinJV grew to consistently higher titres than both reporter viruses, with mean titres of 3.79 × 10^8^, 1 × 10^9^ and 2.35 × 10^9^ FFU/mL at three, five, and seven days, respectively. Comparative statistical analysis between BinJV reporters and WT virus titres was undertaken with a two-way ANOVA with Dunnett’s correction. Both reporters were significantly different on days five (BinJV_zsGreen_; *p* < 0.05, BinJV_mCherry_; *p* < 0.0001) and seven (BinJV_zsGreen_; *p* < 0.001, BinJV_mCherry_; *p* < 0.0001), the BinJV_zsGreen_ virus was the closest in replication kinetics to WT BinJV and was no different in titres at day three. To examine if the BinJV reporters could be quantified live and non-invasively using fluorescence microplate readers, we infected C6/36 cells with serially diluted culture supernatant from the growth kinetics experiment (Figure 2A) in black optical-bottom plates. The fluorescence signals of mCherry and zsGreen were read daily for seven days. Signal was not detected in any reporter virus-infected C6/36 cells until day three, with signal increasing incrementally from day 3 to 5 (data not shown). We noted no cross-reactivity or contamination for either reporter virus, and auto-fluorescence was not present in both mock-infected and BinJV WT-infected C6/36 cells. BinJV reporters gave reliable fluorescence signals (Figure 2B, C) above background fluorescence in mock-infected cells (~0 Fluorescence Units for mCherry and ~0.2 Fluorescence Units for zsGreen). They showed reduced signal in cells infected with serially diluted virus samples. The signal could be reliably detected in cells infected with dilutions 10^−1^ to 10^−3^ for BinJV_mCherry_ and 10^−1^ to 10^−4^ for BinJV_zsGreen_.

Overall, the BinJV_mCherry_ signal units range was the highest in C6/36 cells compared to BinJV_zsGreen_. To examine if it was possible to generate a standard curve for relative fluorescence units and iPA titres, a simple linear relationship between the relative fluorescence units and viral titres generated from iPA-matched samples were plotted and analysed (Figure 2D,E). Three mock-infected wells were also added to calibrate the baseline at 0 FFL/mL. Generally, there was a strong (R^2^ = 0.855) linear relationship between quantitated fluorescence and iPA titres for the BinJV_mCherry_ (Figure 2D) and a more weak but positive general trend (R^2^ = 0.539) for the BinJV_zsGreen_ reporter virus (Figure 2E). The slope for both linear equations was significantly non-zero (BinJV_zsGreen_; *p* = 0.0003, BinJV_mCherry_; <0.0001). This indicates that instead of expensive antibody reagents and scanning equipment used for iPA assay, fluorescence microplate readers may provide a reliable semi-quantifiable alternative to reporter virus titration.

### 3.3. Primary Infection of Mosquito Cells with BinJV_zsGreen_ Inhibits Subsequent Infection with BinJV_mCherry_

To test the utility of BinJV reporters for superinfection exclusion experiments, we first examined the ability of infection with one BinJV to prevent the following infection with another BinJV. The format for superinfection exclusion assays involved a primary infection of C6/36 cells with BinJV_zsGreen_ virus at an MOI = 1 for five days. Mock-infected cells were used as a control. After five days, cells were washed and infected with BinJV_mCherry_ at a MOI = 1 for another five days.

Infection with BinJV_zsGreen_ alone continued throughout all 10 days of the experiment (Figure 3A). Infection with BinJV_mCherry_ without prior infection with BinJV_zsGreen_ also showed a strong increase in mCherry signal over the five days of infection (Figure 3B). In stark contrast, cells previously infected with BinJV_zsGreen_ were completely protected from a secondary infection with BinJV_mCherry_ (Figure 3C), clearly demonstrating a strong interference. No red or green signals were observed in mock controls (Figure 3D).

### 3.4. Primary Infection of Mosquito Cells with BinJV Completely Inhibits Subsequent Infection with ZIKV

Given the previous reports of primary ISF infection leading to restriction of a secondary infection of VIF in mosquito bodies or cells [22,23,24,25], we wanted to examine the utility of our BinJV reporters in the potential exclusion of ZIKV.

We used a ZIKV_mCherry_ reporter virus generated based on a published design [33]. Primary and secondary infections were undertaken with the same conditions as in Figure 3, except for the secondary infections with ZIKV_mCherry_ (MOI = 1). As per the previous experiment, cells were monitored daily using inverted epifluorescence microscopy on green and red channels. Here, primary infections of BinJV_zsGreen_ and mock (media only) secondary infections (Figure 4A), and mock primary infection and secondary infection with ZIKV_mCherry_ (MOI = 1) (Figure 4B) were undertaken as controls. C6/36 cells previously infected with BinJV_zsGreen_ virus completely inhibited a secondary infection with ZIKV_mCherry_ virus (Figure 4C). This was in contrast to mock-infected cells infected with ZIKV_mCherry_, in which a strong increase in mCherry signal was observed from day three to day five after infection (Figure 4B). No red or green signals were observed in mock controls (Figure 4D). 

To validate the epifluorescence imaging results, we harvested supernatants of infected cells for experiments 4B and 4C and infected C6/36 cells quantitating viral titres of culture supernatants with iPA shown in Figure 4B (ZIKV_mCherry_ only) and Figure 4C (Primary infection with BinJV_zsGreen_ and secondary infection with ZIKV_mCherry_). To detect only ZIKV in the BinJV/ZIKV co-infection experiment, a ZIKV-specific anti-E antibody (Z67) was used to titrate the virus on mammalian Vero76 cells (Figure 4E) and mosquito C6/36 cells (Figure 4F). The results confirmed the lack of productive infection with ZIKV_mCherry_ virus in the culture fluid of cells first infected with BinJV_zsGreen_ followed by infection with ZIKV_mCherry_ and the increasing over time viral titres in the culture fluid of cells infected only with ZIKV_mCherry_.

### 3.5. BinJV_zsGreen_ Infection Can Be Visualised In Vivo

Given the success of the visualisation of BinJV reporters in vitro we sought to examine the suitability of the BinJV_zsGreen_ in a preliminary study of intrathoracic injection of adult *Ae. aegypti* mosquitoes. Seven- to ten-day-old mosquitoes were injected with ≈372 FFU of virus per mosquito, and mosquitoes were frozen at seven days post-infection.

We conducted standard frozen sectioning to avoid harsh processing steps used for paraffin embedding. During the frozen sectioning of mosquitoes, the morphology of specimens was disrupted. Despite this, a robust zsGreen signal was detected in most mosquitoes at seven days post-infection (representative mosquito shown in Figure 5A), with the majority of zsGreen signal localised to small areas near the salivary glands (Figure 5B).

Preliminary attempts to examine the zsGreen signal in mosquito samples that were paraffin fixed showed that the zsGreen signal was lost during the procedure with few exceptions (data not shown). While further optimisation is required, these data indicate that BinJV_zsGreen_ reporter can be visualised in vivo.

## 4. Discussion

Here, we report the construction of lineage II ISF BinJV reporter viruses encoding fluorescent proteins zsGreen or mCherry. BinJV reporter viruses provide a powerful tool to study viral infection and pathogenesis of the insect-restricted flaviviruses. Using fluorescence scanning, we demonstrate that BinJV reporter viruses can be semi-quantitated in vitro. We also show that visualisation of BinJV_zsGreen_ is possible in vivo in mosquito tissues.

We show that the BinJV_zsGreen_ reporter viruses induced superinfection exclusion of homologous infections (inhibition of the other BinJV reporter virus) and a heterologous infection, from a ZIKV_mCherry_ reported virus. This demonstrates that infection of *Ae. albopictus* cells with BinJV completely inhibited subsequent infection with a pathogenic VIF, providing strong support for the potential application of insect-specific flaviviruses as biological control agents to inhibit pathogenic virus infections in mosquitoes.

Initial studies on the construction of reporter flaviviruses used various approaches for the insertion of reporter genes, primarily internal ribosome entry site (IRES)-based systems (reviewed by [28]). However, these initial construction methods typically resulted in instability of the reporter genes after serial passaging and virus attenuation due to recombination occurring at the insertion site [28]. Successive rounds of improved method design have led to the capsid duplication strategy, whereby reporter gene stabilisation is attained by introducing the reporter gene between the first 38–50 codons of the duplicated flavivirus capsid gene and a partially or fully codon-optimised complete capsid gene to prevent homology-directed recombination that can hinder efficient reporter virus generation [33,46,47]. However, studies exploiting this strategy have relied on the construction of infectious clones that require propagation in bacteria, increasing the likelihood of mutations compromising reporter gene stability and virus replication. Here, we employed the CPER method combined with the capsid duplication strategy to recover recombinant insect-specific reporter viruses. The CPER-generated viruses retained their reporter genes, evidencing the lack of homology-directed recombination between capsid sequences. Similarly, fluorescence emitted by zsGreen and mCherry proteins were visualised in CPER-transfected cells as early as four days after transfection, indicating successful expression of reporters by these viruses in vitro.

To our knowledge, the semi-quantification of reporter viruses in vitro using microplate readers described here is a reliable quantification strategy for virus titration. It allows the detection and quantification of BinJV without expensive secondary labelling reagents and may be amenable to other virus reporter systems. Additionally, it offers a noninvasive strategy to monitor the dynamics of viral infections in vitro. Finally, it improves the utility of BinJV reporter viruses for fast, high-throughput screening such as those used in whole genome, RNAi-based gene screening to identify viral receptors or antiviral host factors.

While neither reporter virus grew identical to the WT BinJV, the reporter viruses grew to sufficient titres for downstream applications such as superinfection exclusion assays. Superinfection exclusion is commonly observed in virus–virus interactions due to intracellular bottlenecks [48], with multiple reports of superinfection exclusion of flaviviruses in vitro [22,23,24,25]. The current mechanism of flavivirus superinfection exclusion is unknown. Competition for resources or perturbation of the host immune response may be two possibilities [48,49]. Previous studies with a similar regimen of infection with cell fusing agent virus (CFAV) (a lineage one ISF) in C6/36 cells have shown that CFAV decreases but does not entirely restrict a secondary ZIKV infection [23]. Previous studies using superinfection exclusion in DENV isolates in C6/36 cells suggest that exclusion is more significant between more closely related DENV isolates [50]. Given the close genetic relationship between BinJV and ZIKV and the complete inhibition of secondary infection with ZIKV observed here, dISFs may be more suitable for biocontrol. It is also possible that the high replication efficiency of BinJV may account for more efficient inhibition of subsequent infection with ZIKV.

Here, we have demonstrated that primary infection of C6/36 cells with BinJV_zsGreen_ completely inhibits a secondary infection with homologous BinJV_mCherry_ or heterologous ZIKV_mCherry_ using fluorescence microscopy and virus quantitation by iPA. We did not perform a reciprocal experiment with primary infection with BinJV_mCherry_ virus and secondary infection with BinJV_zsGreen_ or an experiment with primary infection with WT BinJV and secondary infections with reporter BinJV or ZIKV viruses. However, data in the literature [22,23,24,25] suggest that these experiments would likely generate results similar to those observed in our experiments.

While a robust signal of reporter genes could be visualised in vitro in C6/36 cells, only a small region of zsGreen fluorescence could be visualised in BinJV_zsGreen_ infected *Ae. aegypti* tissues. The reasons for this are currently unclear. While it is unlikely that BinJV_zsGreen_ has a unique tissue tropism for the salivary gland tissues, it has been previously shown for certain flaviviruses that virus replication is observed preferentially in the proximal lateral and medial lobes of the salivary glands [51,52]. Electron microscopy has also identified the presence of crystalline flavivirus arrays in the salivary glands of vector mosquitoes [53], the formation of which might cause significant zsGreen signal production. Beyond the potential biochemical differences between the salivary glands and other tissues [54], our failure to detect the zsGreen signal might be due to insufficient signal in the zsGreen channel or lower viral titres in other tissues. In naturally blood-fed mosquitoes, there is a temporal tropism of flavivirus infection, with the salivary glands and head being among the last tissues to become infected at 7–10 days [52]. Given that the salivary glands are among the last tissues infected, it is unlikely that BinJV_zsGreen_ has lost the reporter gene as the virus has spread through multiple tissue barriers before entry to the salivary glands. Future work to examine if the lack of reporter gene expression in other tissues is a technical limitation of the system could include detection of BinJV WT or BinJV_zsGreen_ by immunofluorescent staining for a viral protein and cross-referencing it with direct reporter protein fluorescence. In addition, tissue-specific RT-qPCR assays for a BinJV gene or reporter genes could be performed to reveal if these reporter viruses faithfully reproduce their WT BinJV counterpart in vivo.

There has been a recent push toward discovering and characterising ISVs in vector mosquitoes [55,56,57]. However, the interactions between ISV/ISFs and pathogenic arboviruses in the mosquitoes remain poorly defined. We anticipate that the reporter viruses described here will facilitate future superinfection exclusion studies, allowing the investigation of the utility of ISFs as potential biocontrol of pathogenic flaviviruses.

## Figures and Tables

**Figure 1 viruses-14-01501-f001:**
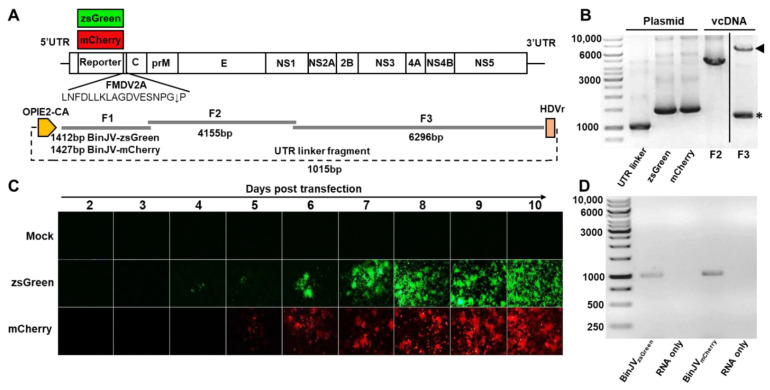
Design and recovery of BinJV reporter viruses. (**A**) Schematics of BinJV and overlapping fragments and linker fragments used for CPER assembly. (**B**) Agarose gel electrophoresis of BinJV fragments amplified from viral cDNA, BinJVzsGreen/mCherry fragments and the UTR linker fragment from plasmids. Arrowhead in F3 lane shows correct size band, while asterisk in F3 lane shows non-specific amplification. The image for F3 is from the same gel with unrelated lanes spliced out for clarity. (**C**) Images of zsGreen and mCherry fluorescence of C6/36 cells transfected with reporter virus CPER were taken at 40× magnification daily from two to ten days post-transfection. (**D**) RT-PCR for the reporter insert of P_1_ BinJV recombinants propagated in C6/36 cells. RNA-only samples refer to the PCR amplification of RNA samples without the reverse transcription step. Amplicons were generated using RGene_F and RGene_R primers binding to the BinJV sequences flanking the reporter genes.

**Figure 2 viruses-14-01501-f002:**
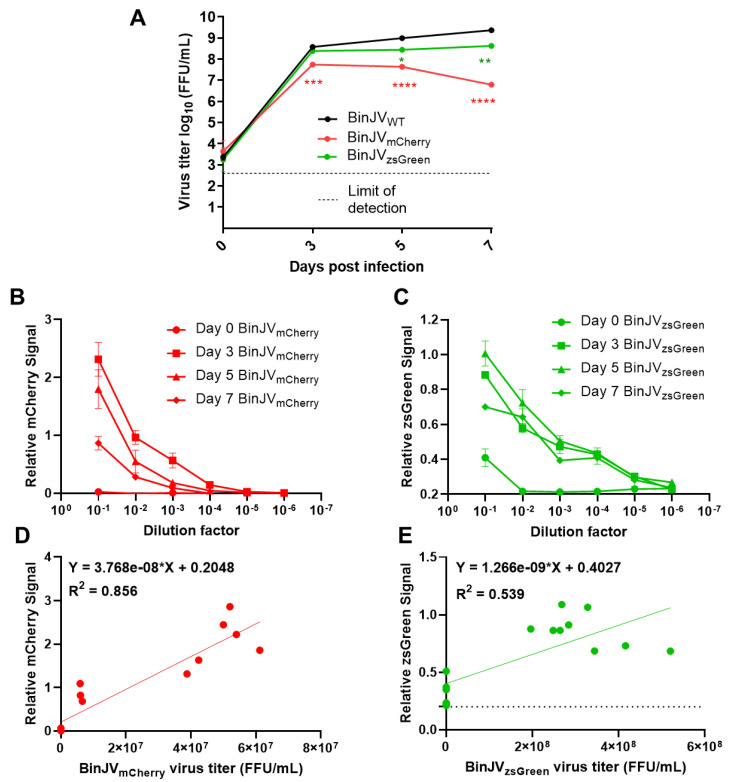
Binjari virus reporters grow to high titres in insect cells and are semi-quantifiable using fluorescent microplate readers. (**A**) Growth kinetics of P_1_ CPER generated BinJV viruses in C6/36 cells infected with WT or reporter BinJV. Cells were infected at MOI = 0.1, culture supernatant were sampled at the indicated time point, and titers were determined using iPA on C6/36 cells. Values are the means from three biological replicates ±SEM. Statistical analysis was a two-way ANOVA with Dunnett’s correction. All comparisons were to WT; * *p* < 0.05, ** *p* < 0.01, *** *p* < 0.001, **** *p* < 0.0001. Relative fluorescence units for serially diluted C6/36 culture supernatant taken from the growth kinetics and reinfected on C6/36 cells showing (**B**) BinJV_mCherry_ signal and (**C**) BinJV_zsGreen_ signal. The linear relationship between matched samples of fluorescence units taken from the 10^−1^ dilution and FFU/mL assay of (**D**) BinJV_mCherry_ (**E**) BinJV_zsGreen_ viruses. For the simple linear regression analysis, 15 matched samples were used. The baseline signal for zsGreen is given as a dotted line. The linear regression function and the coefficient of determination (R^2^) are given on both graphs.

**Figure 3 viruses-14-01501-f003:**
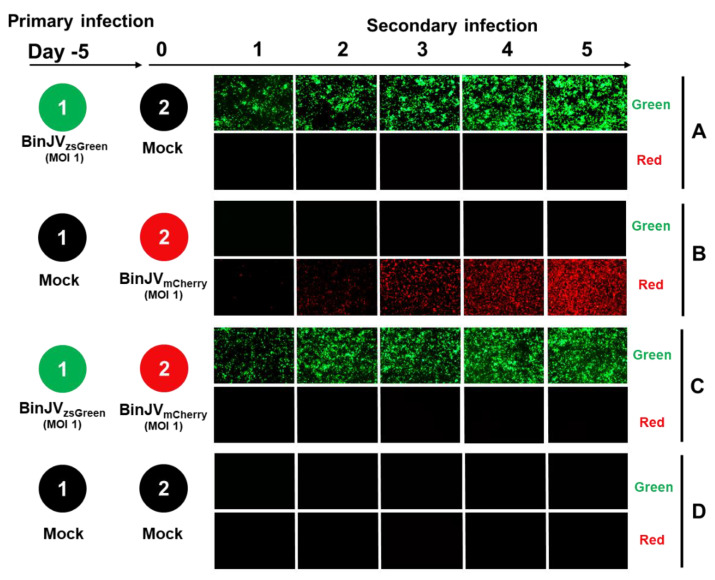
Superinfection exclusion between BinJV_zsGreen_ and BinJV_mCherry_ viruses. Schematic showing the format of the infection trials and representative images of green and red fluorescence channels of C6/36 cells with (**A**) primary infections of BinJV_zsGreen_ at −5 days and mock (media only) secondary infection. (**B**) Mock primary infection at −5 days and BinJV_mCherry_ secondary infection. (**C**) Primary infections of BinJV_zsGreen_ at −5 days, BinJV_mCherry_ secondary infection, and (**D**) mock (uninfected) primary and secondary timepoints.

**Figure 4 viruses-14-01501-f004:**
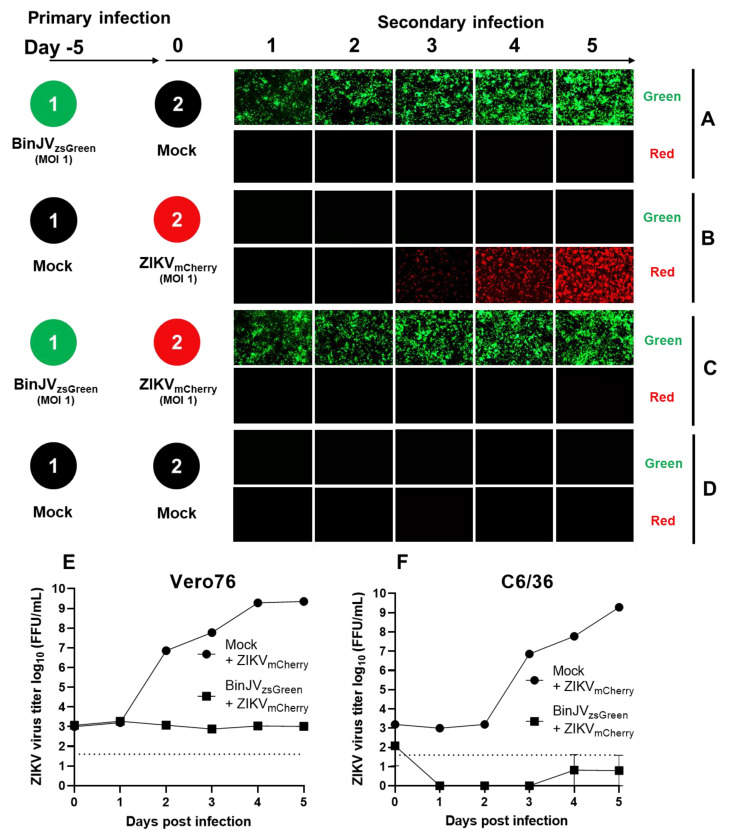
Superinfection exclusion of ZIKV by the insect-specific BinJV. Schematic showing the format of the infection trials and representative images of green and red fluorescence channels of C6/36 cells with (**A**) primary infections of BinJV_zsGreen_ at −5 days and mock (media only) secondary infection. (**B**) Mock primary infection at −5 days and ZIKV_mCherry_ secondary infection. (**C**) Primary infections of BinJV_zsGreen_ at −5 days, ZIKV_mCherry_ secondary infection, and (**D**) mock (uninfected) primary and secondary timepoints. iPA assay of culture supernatant from the indicated time points of experiments (**B**,**C**) titred on Vero76 (**E**) and *Ae. albopictus* (**F**) cells. Values are the means from three replicates ±SEM. The dotted line is the limit of detection.

**Figure 5 viruses-14-01501-f005:**
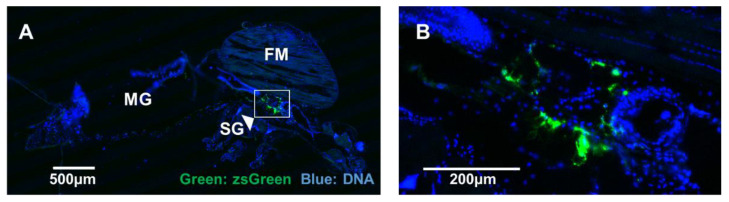
Microinjection of *Aedes aegypti* with BinJV_zsGreen_ reporter virus. (**A**) zsGreen signal (green) in a representative frozen section from a seven-days post-infected mosquito visualised using fluorescence microscopy. DNA was stained to reveal other tissues for orientation. Midguts (MG), flight muscle (FM), and salivary glands (SG) are indicated. (**B**) Magnified region of high signal around the SG. Scale bars are indicated on the figures.

## Data Availability

Raw data underlying figures and uncropped gel photos contained within the article are available on Figshare under https://doi.org/10.6084/m9.figshare.19759930.

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
