# Peer review of "Reporter Flaviviruses as Tools to Demonstrate Homologous and Heterologous Superinfection Exclusion"

_viruses, 2022, doi:10.3390/v14071501_

Round 1

Reviewer 1 Report

Torres et al. detailed the production of reporter BinJVs expressing two types of fluorescence proteins, characterised their growth dynamics, and demonstrated how this technique can be used to view superinfection exclusion against the same virus or a different virus, ZIKV.

This methodology and the superinfection exclusion they demonstrated would be of immediate interest to those studying virus-virus interaction, particularly in vector biology. 

The only critical comment I have is for line 307, which suggests that Fig. 1D should show iPA results instead of an electrophoresis gel.

I have one question out of curiosity. The authors may decide themselves if changes in the manuscript are warranted. In Fig. 1D, the authors used RT-PCR to detect the presence of reporter genes in P1 viruses and concluded that the inserts were stably retained. What RT-PCR primers were used? More specifically, do they bind to the reporter genes or to the virus genome regions flanking the inserts?

Author Response

Author responses in bold.

Torres et al. detailed the production of reporter BinJVs expressing two types of fluorescence proteins, characterised their growth dynamics, and demonstrated how this technique can be used to view superinfection exclusion against the same virus or a different virus, ZIKV.

This methodology and the superinfection exclusion they demonstrated would be of immediate interest to those studying virus-virus interaction, particularly in vector biology. 

The only critical comment I have is for line 307, which suggests that Fig. 1D should show iPA results instead of an electrophoresis gel.

We appreciate the error spotted; this was from a previous draft of the manuscript; the sentence has been deleted.

I have one question out of curiosity. The authors may decide themselves if changes in the manuscript are warranted. In Fig. 1D, the authors used RT-PCR to detect the presence of reporter genes in P1 viruses and concluded that the inserts were stably retained. What RT-PCR primers were used? More specifically, do they bind to the reporter genes or to the virus genome regions flanking the inserts?

The primers RGene_F/R bind to the sequences flanking the inserts. We have added the line “Amplicons generated using RGene_F and RGene_R primers binding to the BinJV sequences flanking the reporter genes.” in the Fig. 1 legend.

Reviewer 2 Report

In this investigation, Torres et al. developed a lineage II insect specific favivirus, BinJV reporter viruses that express fluorescent proteins zsGreen or mCherry. It could be cultivated in a mosquito cell line, semi-quantitated live in vitro using a plate reader, and visually inspected in a frozen section of Aedes aegypti Because BinJV belongs to the lineage II insect-specific flaviviruses, the fluorescent expressing BinJV reporter virus might be used to study flavivirus superinfection in vivo and in vitro. The author discovered that initial BinJVzsGreen infection of C6/36 cells totally blocks both homologous BinJVmCherry and heterologous ZIKVmCherry secondary infection, implying insect-specific flaviviruses' suitability for biocontrol agents to inhibit pathogenic virus in mosquitoes.

Has the author tried the-primary infection with BinJVmCherry followed by BinJVzsGreen for the superinfection experiment? I'm curious if the infectivity of reporter viruses is affected by different tagged fluorescent proteins. Switching primary and secondary infections of the BinJv producing different fluorescent proteins can demonstrate this. Also, did the author use RT-PCR to detect the secondary reporter virus, either homologous or heterologous, to ensure that the lack of red signal is due to no virus and not because the green signal obscures the red signal?

Regarding the distribution of the green signal in the mosquito frozen section restricted only in salivary glands at 7 dpi, it seems that the BinJVzsGreen has low infectivity or propagation in mosquitoes. The author may provide an explanation or discussion for this scenario.

Figure 1B: The additive lane “F3” from a different gel should be spliced out.

Figure 1D: It seems the author presented misinformation in Figure 1. There is no iPA result in Fig 1D as mentioned on lines 181 and 307.

Why did the author use different antibodies for titration of Zik virus (anti ZIKV E protein Z67) and P1ZIKVmCherry (anti-NS1 flavivirus 4G4)?

Minor corrections:

1.       There should be a space between the number and degree symbol, on lines; 111, 221, 230.

2.       It seems reference number 35 is missing.

3.       Add relative humidity after 75% on line 230

4.       Correct experiments A) and C) to B and C) on line 404.

5.   Remove USA on lines 102, 121, and 191.

Author Response

Author responses in bold.

In this investigation, Torres et al. developed a lineage II insect specific favivirus, BinJV reporter viruses that express fluorescent proteins zsGreen or mCherry. It could be cultivated in a mosquito cell line, semi-quantitated live in vitro using a plate reader, and visually inspected in a frozen section of Aedes aegypti Because BinJV belongs to the lineage II insect-specific flaviviruses, the fluorescent expressing BinJV reporter virus might be used to study flavivirus superinfection in vivo and in vitro. The author discovered that initial BinJVzsGreen infection of C6/36 cells totally blocks both homologous BinJVmCherry and heterologous ZIKVmCherry secondary infection, implying insect-specific flaviviruses' suitability for biocontrol agents to inhibit pathogenic virus in mosquitoes.

Has the author tried the-primary infection with BinJVmCherry followed by BinJVzsGreen for the superinfection experiment? I'm curious if the infectivity of reporter viruses is affected by different tagged fluorescent proteins. Switching primary and secondary infections of the BinJv producing different fluorescent proteins can demonstrate this. Also, did the author use RT-PCR to detect the secondary reporter virus, either homologous or heterologous, to ensure that the lack of red signal is due to no virus and not because the green signal obscures the red signal?

We have not undertaken a primary infection with BinJVmCherry followed by BinJVzsGreen for superinfection exclusion. Homologous superinfection exclusion is widely reported for Flaviviruses and we don’t believe that it is necessary to re-demonstrate this effect. It is also highly unlikely that the phenotype may be due to mCherry or zsGreen specific signals as we clearly show that both are readily detected. Given that the excitation and emission spectra are suitably different between zsGreen and mCherry there is very little possibility that fluorescence signal with one will “obscure” the other when using specific filters.

We have tested for the presence of secondary reporter virus in the culture fluid (Fig 4E and F) and found no evidence of productive virus replication. We, therefore, do not believe that additional RT-PCR data are required.

Regarding the distribution of the green signal in the mosquito frozen section restricted only in salivary glands at 7 dpi, it seems that the BinJVzsGreen has low infectivity or propagation in mosquitoes. The author may provide an explanation or discussion for this scenario.

A paragraph discussing the observation of zsGreen signal in salivary glands only in infected mosquitoes has been added to the discussion (Lines: 492-508). We discuss the possibilities of tissue tropism or potential technical limitations of the system. 

Figure 1B: The additive lane “F3” from a different gel should be spliced out.

Apologies, this is actually from the same gel but with other lanes of the gel spliced out of the image. Undoctored and annotated gel photos have been uploaded as part of the peer-review process and are available on the Figshare DOI: 10.6084/m9.figshare.19759930. A line has been added to separate the two gel pictures in the figure and the corresponding figure legend has been amended to: “The image for F3 is from the same gel with unrelated lanes spliced out for clarity”

Figure 1D: It seems the author presented misinformation in Figure 1. There is no iPA result in Fig 1D as mentioned on lines 181 and 307.

We appreciate the errors spotted with this figure; this was from a previous draft of the manuscript. The sentence and detail have been deleted.

Why did the author use different antibodies for titration of Zik virus (anti ZIKV E protein Z67) and P1ZIKVmCherry (anti-NS1 flavivirus 4G4)?

Quantification of BinJV and ZIKV virus stocks by iPA can be performed using anti-pan-flavivirus antibodies (anti-NS1 4G4 antibody). However, given that iPA quantification of superinfection exclusion samples required an antibody that specifically discriminates between ZIKV and BinJV, a ZIKV-specific Z67 antibody was chosen for this purpose. We have added a line to make this clearer in the text (Line 412-415).

Minor corrections:

  1. There should be a space between the number and degree symbol, on lines; 111, 221, 230.

This has been changed.

  1. It seems reference number 35 is missing.

Apologies this ghost citation has been removed from the document.

  1. Add relative humidity after 75% on line 230

This has been added.

  1. Correct experiments A) and C) to B and C) on line 404.

This has been changed.

  1.  Remove USA on lines 102, 121, and 191.

This has been changed.

Round 2

Reviewer 2 Report

Regarding the author's vehemently claimed the construction of reporter viruses for lineage II insect-specific flaviviruses to use as a tractable system for exploring flavivirus superinfection exclusion in vitro and in vivo. By utilizing these reporter viruses, the authors showed that the BinJVzsGreen reporter viruses induced superinfection exclusion of homologous infections and a heterologous infection from a ZIKVmCherry reported virus, implying that infection of Ae. albopictus cells with BinJV completely inhibited subsequent infection with a pathogenic VIF. There are two major points that should be strongly supported for publication:

1. The author may need to show that the fluorescent-tagged viruses can be seen in the tissue section of Ae. aegypti in a way that is comparable to the wild-type virus infected tissue detected by immunofluorescent staining if the author wishes to assert the fluorescent-tagged viruses' capacity as reporters in vivo. Otherwise, this power was exaggerated.

2. For the superinfection experiment, it was not a good experimental design to perform the first infection with solely the zsGreen-tagged virus, followed by the mCherry-tagged virus, without verifying the outcome by switching infection sequences. The experiment could need to be demonstrated either by swapping reporter viruses for infection or by primary infection with the wild-type virus and then the tagged viruses later.

Author Response

Author responses are in blue

  1. The author may need to show that the fluorescent-tagged viruses can be seen in the tissue section of Ae. aegyptiin a way that is comparable to the wild-type virus infected tissue detected by immunofluorescent staining if the author wishes to assert the fluorescent-tagged viruses' capacity as reporters in vivo. Otherwise, this power was exaggerated.

We appreciate that our ability to detect BinJV zsGreen signal in only one tissue type may be a technical limitation. However, we believe we have been honest in describing this in the abstract and believe that we do show that the viruses can be visualised in vivo. “At seven days post-infection, a strong fluorescence in the vicinity of salivary glands was detected in frozen sections.” We have also added in the Discussion potential experiments to overcome technical limitations of the BinJV reporter detection in response to the reviewer’s comments  (Lines 517-521).

  1. For the superinfection experiment, it was not a good experimental design to perform the first infection with solely the zsGreen-tagged virus, followed by the mCherry-tagged virus, without verifying the outcome by switching infection sequences. The experiment could need to be demonstrated either by swapping reporter viruses for infection or by primary infection with the wild-type virus and then the tagged viruses later.

We appreciate that there are many ways to design and test the superinfection exclusion phenomenon. We have now stated in the Discussion that we did not perform the experiments suggested by the reviewer and cited publications that, in our opinion, would suggest that these experiments would likely yield results similar to those observed in our experiments (Lines 493-500). We contend that we have been honest through the text with the exact nature of the experiments that were conducted and also clearly state this in the abstract: “We demonstrate that primary infection of C6/36 cells with BinJVzsGreen completely inhibits a secondary infection with homologous BinJVmCherry or heterologous ZIKVmCherry using fluorescence microscopy and virus quantitation by iPA.”